# SCALING FP8 TRAINING TO TRILLION-TOKEN LLMS

**Maxim Fishman** [†*]  **Brian Chmiel** [†*]  **Ron Banner** [†]   **Daniel Soudry** [°]

[†]Intel, Israel
[°]Department of Electrical and Computer Engineering - Technion, Haifa, Israel

{maxim.fishman, brian.chmiel, ron.banner}@intel.com
{daniel.soudry}@gmail.com

## ABSTRACT

We train, for the first time, large language models using FP8 precision on datasets up to 2 trillion tokens — a 20-fold increase over previous limits. Through these extended training runs, we uncover critical instabilities in FP8 training that were not observable in earlier works with shorter durations. We trace these instabilities to outlier amplification by the SwiGLU activation function. Interestingly, we show, both analytically and empirically, that this amplification happens only over prolonged training periods, and link it to a SwiGLU weight alignment process. To address this newly identified issue, we introduce Smooth-SwiGLU, a novel modification that ensures stable FP8 training without altering function behavior. We also demonstrate, for the first time, FP8 quantization of both Adam optimizer moments. Combining these innovations, we successfully train a 7B parameter model using FP8 precision on 256 Intel Gaudi2 accelerators, achieving on-par results with the BF16 baseline while delivering up to a $\sim 34\%$ throughput improvement. A reference implementation is supplied in https://github.com/Anonymous1252022/Megatron-DeepSpeed.

## 1 INTRODUCTION

Large Language Models (LLMs) have revolutionized natural language processing, demonstrating remarkable capabilities across a wide range of tasks. However, the computational demands of training these models have become increasingly challenging, driving the need for more efficient training methods. Low precision formats, particularly FP8, have emerged as a promising solution to reduce memory usage and accelerate training. Recent work by Peng et al. (2023) has demonstrated the potential of FP8 training for LLMs. However, these studies have been limited to datasets of up to 100 billion tokens, leaving open questions about the scalability and stability of FP8 in truly large-scale training scenarios.

In this paper, we present advancements in FP8 training for LLMs, successfully scaling to datasets of up to 2 trillion tokens — a 20-fold increase over previous limits. This leap in scale has revealed critical instabilities in FP8 training that were not observable in earlier, shorter-duration studies. Through rigorous analysis, we trace these instabilities to a previously unidentified phenomenon: the amplification of outliers by the SwiGLU activation function (Shazeer (2020a)), which becomes pronounced only after extended training periods. The severity of this issue is illustrated in Fig. 2(a), where we present the training loss of Llama2 7B using FP8 precision. The graph clearly shows a dramatic divergence caused by outliers after processing 220B tokens - twice the dataset size explored in previous work (Peng et al. (2023)).

To address this newly discovered challenge, we introduce Smooth-SwiGLU, a novel modification to the standard SwiGLU activation that effectively mitigates outlier amplification without altering the function's behavior. This innovation ensures stable FP8 training across extended durations, enabling the use of FP8 precision in large-scale LLM training without compromising model performance.

---

[*]Equal contribution.

Furthermore, we push the boundaries of low-precision optimization by demonstrating, for the first time, the successful quantization of both Adam optimizer moments to FP8. This advancement reduces memory usage during training, further enhancing the efficiency of large-scale LLM development.

By combining these innovations - Smooth-SwiGLU and FP8 quantization of optimizer moments - we successfully train a 7B parameter model using FP8 precision on 256 Intel Gaudi2 accelerators. Our approach achieves results on par with the BF16 baseline while delivering up to 34% throughput improvement, marking a significant leap in training efficiency for large-scale language models.

Our paper makes several key contributions:

- We demonstrate the first successful FP8 training of LLMs on datasets up to 2 trillion tokens, far surpassing previous limits and revealing critical instabilities in extended training regimes.
- We identify the root cause of these instabilities: outlier amplification by the SwiGLU activation function over prolonged training periods.
- We link, analytically and empirically, the outlier amplification to a weight alignment happening during training with $\ell_2$ regularization, in the SwiGLUs with sufficiently large inputs.
- We introduce Smooth-SwiGLU, a novel activation function that ensures stable FP8 training without altering model behavior, enabling efficient large-scale LLM training.
- We present the first implementation of FP8 quantization for both Adam optimizer moments, further optimizing memory usage in LLM training.
- We achieve on-par results with BF16 baselines on downstream tasks while providing throughput improvements on Intel Gaudi2 accelerators, demonstrating the practical viability of our approach for state-of-the-art LLM training.

## 2 BACKGROUND AND CHALLENGES OF FP8 TRAINING IN LLMs

The computational demands of Large Language Models (LLMs) have driven a shift from traditional FP32 to reduced-precision formats. While FP16 and BF16 have become standard in many training tasks (Micikevicius et al., 2017; Scao et al., 2022; Smith et al., 2022), FP8 represents the next step in this progression towards lower precision. Micikevicius et al. (2022) standardized two FP8 formats for deep learning: E4M3 (4 exponent bits, 3 mantissa bits) optimized for weights and activations, and E5M2 (5 exponent bits, 2 mantissa bits) suitable for gradients.

FP8 shows promise for large-scale training, especially with support from modern hardware like NVIDIA's H100 and Intel's Gaudi2. However, its limited range necessitates careful scaling techniques to maintain numerical stability and model performance.

The primary challenge in FP8 training for LLMs stems from its limited dynamic range. To address this, researchers have developed various scaling techniques. Global loss scaling (Micikevicius et al., 2017) multiplies the entire loss by a constant factor to prevent gradient underflow during backpropagation. Per-tensor scaling (Sun et al., 2019) takes a more granular approach, scaling each tensor individually based on its specific range of values. These techniques allow for better utilization of the FP8 format's limited range. However, (Lee et al., 2024) demonstrates that, even with these techniques, FP8 training can lead to significant instabilities, highlighting the ongoing challenges in reduced-precision LLM training.

Implementation of these scaling techniques typically follows one of two approaches: just-in-time scaling or delayed scaling. Just-in-time scaling dynamically adjusts scaling factors based on the current data distribution. However, it often proves impractical in FP8 training due to the need for multiple data passes, which can negate the performance benefits of using FP8. Delayed scaling, on the other hand, selects scaling factors based on data distributions from preceding iterations. While more practical, it assumes consistent statistical properties across iterations, making it vulnerable to outliers that can disrupt this consistency and potentially destabilize the training process.

Recent work by Peng et al. (2023) demonstrated the first empirical results of training LLMs using FP8 format up to 100 billion tokens, validating the potential of FP8 for large-scale training, but less applicable in real-world scenarios that usually require larger training. Our research extends this work, successfully scaling FP8 training to datasets of up to 2 trillion tokens. We introduce novel techniques

to address the challenges of FP8's limited dynamic range and the instabilities that emerge in extended training scenarios. Our approach not only overcomes these limitations but also achieves substantial improvements in memory usage and training speed, demonstrating the viability of FP8 training for truly large-scale LLM development.

# 3 OUTLIER AMPLIFICATION IN LARGE-SCALE FP8 TRAINING

The presence of outliers has been observed in numerous studies (Yang et al., 2024; Bondarenko et al., 2023), particularly in the activations during inference. One of the previously solution presented to confront with these outliers for inference, is to apply rotation to the activations (Liu et al., 2024). These outliers can significantly impact the stability and performance of the model, as they introduce extreme values that are difficult to manage within the limited dynamic range of reduced-precision formats like FP8. Our work reveals that these outliers become particularly prominent in the later stages of training large language models (LLMs) with large-scale datasets.

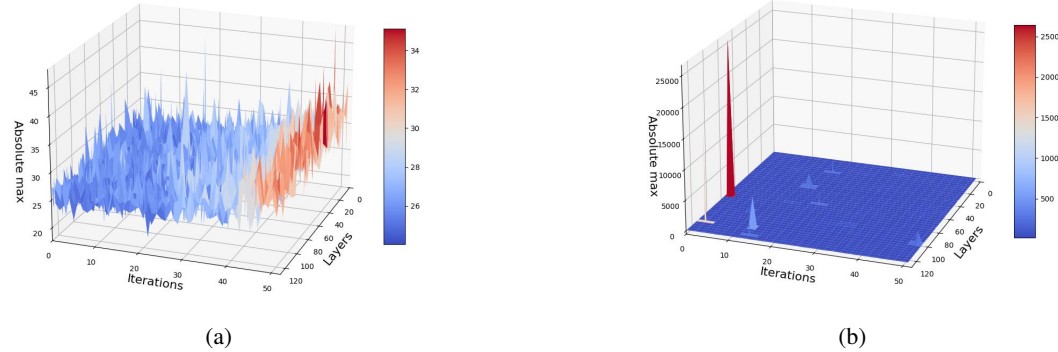

Figure 1: Comparison of activation maximum values across different layers during 50 iterations of training: (a) At the beginning of training, showing stable maximum values. (b) After 200B tokens of training, revealing sporadic but significant outliers (notice the change in the z-axis scale).

As shown in Fig. 1, these outliers emerge only after processing approximately 200 billion tokens during training. This phenomenon poses significant challenges to maintaining numerical stability and model performance, especially when using methods that assume consistency across iterations. The sudden appearance of these outliers, which are crucial for model performance (Sun et al., 2024), disrupts the statistical assumptions underlying FP8 training techniques, potentially leading to instability or divergence in the training process.

The emergence of these outliers in the later stages of training is particularly problematic for FP8 formats due to their limited dynamic range. Unlike higher precision formats such as FP32 or even BF16, FP8 has a much narrower range of representable values. When outliers exceed this range, they can cause overflow or underflow, leading to a loss of critical information and potentially destabilizing the entire training process.

Moreover, the sporadic nature of these outliers, as evident in Fig. 1b, makes them challenging to predict and manage. Traditional scaling techniques, which rely on consistent statistical properties across iterations, struggle to adapt to these sudden, extreme values. This unpredictability further complicates the task of maintaining numerical stability in FP8 training, especially as the scale of the dataset and the duration of training increase.

# 4 SWIGLU AND OUTLIER AMPLIFICATION

While the previous section highlighted the general problem of outlier emergence in large-scale FP8 training, our investigation reveals that the SwiGLU (Swish Gated Linear Unit) activation function plays a crucial role in amplifying these outliers. This section explores the structure of SwiGLU and demonstrates how its unique properties contribute to the generation and amplification of outliers.

## 4.1 SWIGLU STRUCTURE

The Transformer architecture (Vaswani et al., 2017), which forms the foundation of modern LLMs, has undergone several modifications to enhance performance and efficiency. One notable example

is the inclusion of the SwiGLU (Swish Gated Linear Unit) (Shazeer, 2020b) activation function in models like LLaMA (Touvron et al., 2023) and PaLM (Chowdhery et al., 2022).

Let $\mathbf{x} \in \mathbb{R}^d$ be the input vector from the previous layer. For two weight vectors $\mathbf{w}_1, \mathbf{w}_2 \in \mathbb{R}^d$, the SwiGLU neuron is defined as follows

$$\text{SwiGLU}_{\mathbf{w}_1,\mathbf{w}_2}(\mathbf{x}) \triangleq (\mathbf{x}^\top \mathbf{w}_1)\text{Swish}(\mathbf{x}^\top \mathbf{w}_2) \triangleq (\mathbf{x}^\top \mathbf{w}_1)(\mathbf{x}^\top \mathbf{w}_2)\sigma(\mathbf{x}^\top \mathbf{w}_2),$$

where $\sigma(z) \triangleq 1/(1 + e^{-z})$ is the sigmoid function.

While other standard neuron types, such ReLU, GeLU, and Swish at most linear at large input magnitudes (i.e., $\lim_{u \to \pm\infty} |f(u)/u| \leq 1$), the SwiGLU activation is quadratic and can reach much larger values (and cause very strong outliers) if $\mathbf{w}_1$ and $\mathbf{w}_2$ are sufficiently aligned (e.g., if $\mathbf{w}_1 = \mathbf{w}_2$ and $\mathbf{w}_1^\top \mathbf{x} = 1$ then $\lim_{c \to \infty} \text{SwiGLU}_{\mathbf{w}_1,\mathbf{w}_2}(c\mathbf{x})/c^2 = 1$). As we show next, precisely such alignment happens during training for neurons with sufficiently large inputs.

## 4.2 THEORETICAL ANALYSIS OF WEIGHT CORRELATION IN SWIGLU

Next, we analyze the behavior of the SwiGLU neuron during training and show its weight vectors tend to align perfectly if the magnitude of its input increases above some threshold. This causes the SwiGLU output magnitude to increase significantly during training, potentially resulting in outliers.

To show this, we assume the SwiGLU neuron is embedded in a neural network with $k$ parameters. The rest of the parameters in the network are denoted by $\theta \in \mathbb{R}^{k-2d}$. We train the neural network with some $\ell_2$ regularization

$$\min_{\mathbf{w}_1,\mathbf{w}_2,\theta} \sum_{n=1}^{N} \ell_n \left(\text{SwiGLU}_{\mathbf{w}_1,\mathbf{w}_2}\left(\mathbf{x}_n\left(\theta\right)\right),\theta\right) + \frac{\mu}{2}\left(\|\mathbf{w}_1\|^2 + \|\mathbf{w}_2\|^2 + \|\theta\|^2\right), \tag{1}$$

where $\mu > 0$ is regularization strength, $N$ is the number of training samples, and $\ell_n(u_n, \theta)$ is the per-sample loss as a function of the SwiGLU output and the rest of the neural network parameters. We find that

**Theorem 1.** Suppose we converge to a stationary point $(\mathbf{w}_1, \mathbf{w}_2, \theta)$ of the loss function, and for all samples $n$, $\sigma'(\mathbf{x}_n^\top(\theta)\mathbf{w}_2) \to 0$. Then, at this stationary point, $\mathbf{w}_1 \to \mathbf{w}_2$ or $\mathbf{w}_1 \to -\mathbf{w}_2$.

**Proof.** At a stationary point $(\mathbf{w}_1, \mathbf{w}_2, \theta)$ we have $\forall i \in \{1, 2\}$:

$$\sum_{n=1}^{N} \nabla_{\mathbf{w}_i} \ell_n(\text{SwiGLU}_{\mathbf{w}_1,\mathbf{w}_2}(\mathbf{x}_n(\theta)),\theta) + \mu\mathbf{w}_i = 0.$$

Using the chain rule we obtain the following two equations

$$0 = \sum_n \mathbf{x}_n\mathbf{x}_n^\top \mathbf{w}_2 \sigma(\mathbf{w}_2^\top \mathbf{x}_n)\delta_n + \mu\mathbf{w}_1$$

$$0 = \sum_n \mathbf{x}_n\mathbf{x}_n^\top \mathbf{w}_1 \left(\sigma(\mathbf{w}_2^\top \mathbf{x}_n) + (\mathbf{x}_n^\top \mathbf{w}_2)\sigma'(\mathbf{x}_n^\top \mathbf{w}_2)\right)\delta_n + \mu\mathbf{w}_2,$$

where we defined $\delta_n(\mathbf{w}_1, \mathbf{w}_2, \theta) \triangleq \left.\frac{\partial \ell_n(u_n,\theta)}{\partial u_n}\right|_{u_n=\text{SwiGLU}_{\mathbf{w}_1,\mathbf{w}_2}(\mathbf{x}_n(\theta))}$ and with a slight abuse of notation, we suppressed the dependence of $(\mathbf{w}_1, \mathbf{w}_2, \theta)$ and $\mathbf{x}_n(\theta)$ on $\theta$. Given the assumption

$$\forall n : \sigma'(\mathbf{x}_n^\top \mathbf{w}_2) \to 0$$

at this limit we obtain

$$0 = \sum_n \mathbf{x}_n\mathbf{x}_n^\top \mathbf{w}_2 \sigma(\mathbf{w}_2^\top \mathbf{x}_n)\delta_n + \mu\mathbf{w}_1 \quad ; \quad 0 = \sum_n \mathbf{x}_n\mathbf{x}_n^\top \mathbf{w}_1 \sigma(\mathbf{w}_2^\top \mathbf{x}_n)\delta_n + \mu\mathbf{w}_2.$$

Now, defining $\lambda_n = -\mu^{-1}\delta_n\sigma(\mathbf{w}_2^\top \mathbf{x}_n)$, the above equations become

$$\mathbf{w}_1 = \sum_n \lambda_n\mathbf{x}_n\mathbf{x}_n^\top \mathbf{w}_2 = A\mathbf{w}_2 \quad ; \quad \mathbf{w}_2 = \sum_n \lambda_n\mathbf{x}_n\mathbf{x}_n^\top \mathbf{w}_1 = A\mathbf{w}_1.$$

where $A = \sum_n \lambda_n \mathbf{x}_n \mathbf{x}_n^\top$ is a symmetric matrix. This implies

$$\mathbf{w}_1 = A\mathbf{w}_2 = A^2\mathbf{w}_1 \quad ; \quad \mathbf{w}_2 = A\mathbf{w}_1 = A^2\mathbf{w}_2 . \qquad (2)$$

Since $A$ is symmetric this implies that both $\mathbf{w}_1$ and $\mathbf{w}_2$ are eigenvectors of $A$ with the same eigenvalue: 1 or $-1$. Plugging this into equation 2 we obtain $\mathbf{w}_1 = \mathbf{w}_2$ or $\mathbf{w}_1 = -\mathbf{w}_2$. ∎

Note this result holds also if we replace the swish activation $\sigma$ in SwiGLU with other GLU variants (Shazeer, 2020a), since we did not use any specific properties of the Swish. Thus, practically, with regularization and sufficient training, the weight vectors $\mathbf{w}_1$ and $\mathbf{w}_2$ must converge to either identical or opposite directions, i.e., $\mathbf{w}_1 \approx \mathbf{w}_2$ or $\mathbf{w}_1 \approx -\mathbf{w}_2$ — if $\sigma'(\mathbf{x}_n^\top (\theta) \mathbf{w}_2)$ is typically small. Since $\sigma'(z)$ decays exponentially fast as $|z|$ increases, this simply means that the neuron inputs are typically not too small. This can happen in the case in which $\|\mathbf{w}_2\|$ is sufficiently large, and typically $\left|\mathbf{w}_2^\top \mathbf{x}_n(\theta)\right| > 0$. One should expect the latter condition to be the generic case when we fit a neural network to a large dataset of size $N$, where $N \gg k$ (i.e., we are in an under-parameterized regime) and zero loss is not reachable—since then the neural network does not have spare capacity to set specific neuron inputs to zero (in addition to fitting the data). And indeed, we observe (Fig. 9 in the Appendix) that after training $|\mathbf{w}_2^\top \mathbf{x}_n| > 1$ for $\sim 99\%$ of the tokens, in the outlier neuron. Interestingly, this alignment phenomenon occurs due to $\ell_2$ regularization, even if it is very weak. In fact, weak regularization will lead to larger weight norms, strengthening this effect.

## 4.3 OBSERVING WEIGHT CORRELATION GROWTH AND TRAINING INSTABILITY

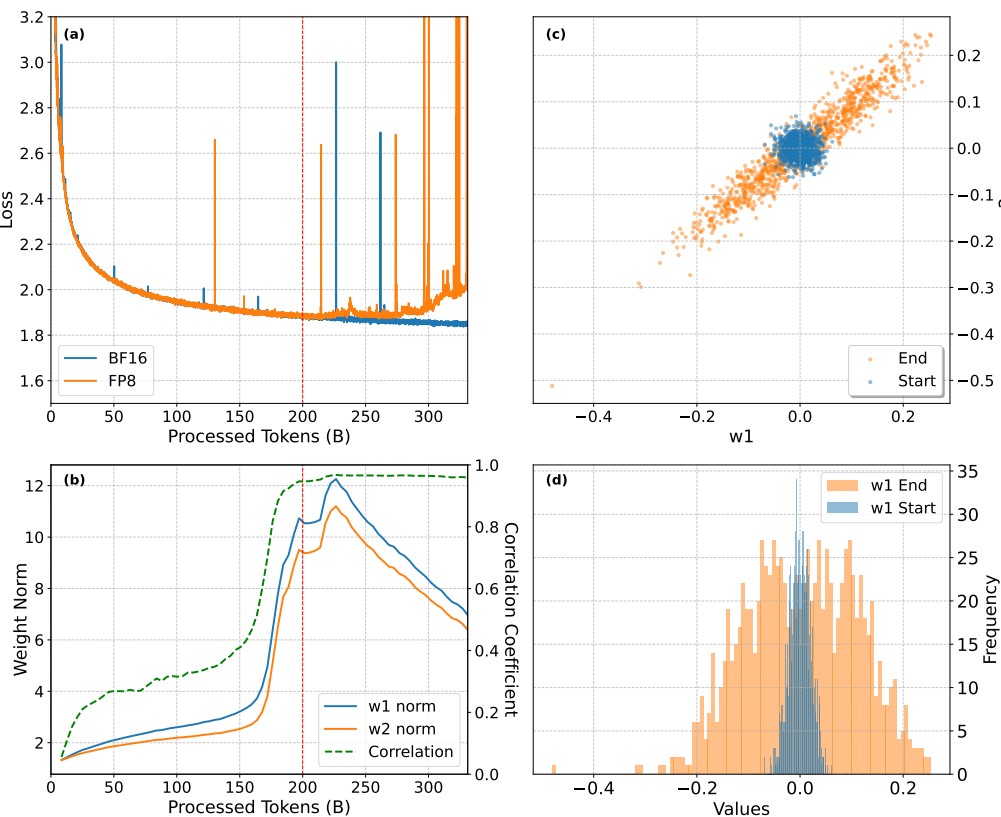

Figure 2: **(a):** Training loss of LlaMA2-7b with BF16 and FP8 precision, where a significant loss divergence is seen for FP8 after step $\sim 200$B tokens. **(b):** Dynamics of the $\mathbf{w}_1$ and $\mathbf{w}_2$ norms, and their correlation during training, for a specific channel that generates outliers. A drastic increase in correlation and norm is observed at the same point where we start to see loss degradation in (a). **(c):** Scatter plot of an outlier channel elements in $\mathbf{w}_1$ and $\mathbf{w}_2$, at an early training stage (8B tokens) and late training stage (330B tokens), demonstrating minimal correlation at start of the training and high correlation in the later stage. **(d):** Histogram of an outlier channel of $\mathbf{w}_1$ at an early training stage (8B tokens) and late training stage (330B tokens).

In our experiments, we observed a clear relationship between the increasing correlation of the weight matrices $\mathbf{w}_1$ and $\mathbf{w}_2$ and the eventual divergence in FP8 training loss.

In Fig. 2 we see the process of weight alignment and its impact on FP8 training develops precisely as our theory predicts. As training progresses, $\|\mathbf{w}_2\|$ in the outlier channel grows, eventually exceeding a critical threshold satisfying our Theorem's assumption. Thus, the weight vectors $\mathbf{w}_1$ and $\mathbf{w}_2$ in these channels begin to align rapidly—i.e. the correlation between $\mathbf{w}_1$ and $\mathbf{w}_2$ is initially low, but then it increases drastically between 125B and 210B tokens. Interestingly, it seems this alignment happens simultaneously with further norm growth. This combination of high correlation and increased weight norm creates ideal conditions for generating extreme activation values, or "spikes."

These activation spikes, in turn, lead to the divergence of FP8 training, as shown in Fig. 2a. Importantly, while we primarily observe strong positive correlations in this example, our theory also predicts the possibility of strong negative correlations. We also observe these, as can be seen in Fig. 7 in the Appendix.

This phenomenon highlights the unique challenges posed by SwiGLU in FP8 training of large language models. The gradual alignment of weights, combined with norm growth, creates a scenario where outliers become increasingly likely and severe as training progresses. Consequently, instabilities may not be apparent in shorter training runs but emerge as critical issues in large-scale, long-duration training scenarios. This explains why such problems have not been observed in previous, smaller-scale studies of FP8 training.

## 4.4 SMOOTH-SWIGLU

As demonstrated earlier, the SwiGLU activation function can lead to outliers in the input of the last linear layer of the MLP component. These outliers pose a significant challenge when using FP8 precision with delayed scaling, which relies on the assumption that statistical properties remain consistent across layers. The sudden spike in value caused by SwiGLU disrupts this continuity, leading to instability in the training process. In Fig. 3 we demonstrate that disabling the quantization of the last linear layer in the MLP component (output of SwiGLU) allows Llama2 FP8 to successfully converge with large datasets, addressing the previously observed divergence issues. This shows that other components in Llama architecture, such as RMS Norm or MHA are not the cause of instability in FP8 training.

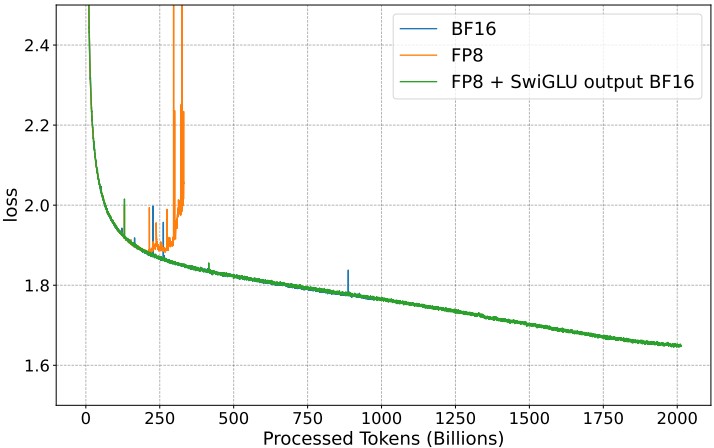

Figure 3: Training loss of Llama2 FP8 with and without quantization of SwiGLU output. As can be seen the cause of the divergence of standard FP8 is the amplification of the SwiGLU (input to $\mathbf{w}_3$).

While disabling quantization of the SwiGLU output effectively prevents divergence, it reduces the potential acceleration benefits of FP8. To maintain full FP8 acceleration while addressing the outlier problem, we propose a novel modification called Smooth-SwiGLU. Figure 4 illustrates the key idea

behind Smooth-SwiGLU: applying a scaling factor to the linear branch of the SwiGLU function and rescaling it back after the last linear layer. This approach prevents outliers in the quantization of the input to the last linear layer while preserving the overall function of the SwiGLU activation, enabling us to fully leverage FP8 precision throughout the network.

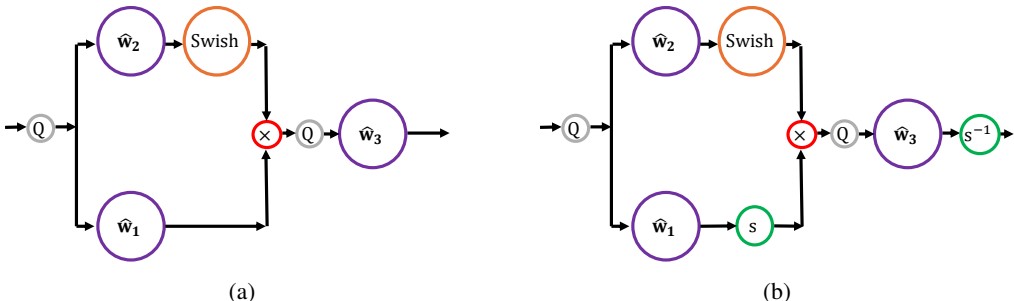

(a)  (b)

Figure 4: A standard quantized MLP component containing the original quantized SwiGLU **(a)** and the proposed quantized Smooth-SwiGLU **(b)**, which improves the stability under FP8 training. Here, $s$ is the scaling factor, $\hat{\mathbf{w}}_1, \hat{\mathbf{w}}_2$ and $\hat{\mathbf{w}}_3$ are the quantized weights, and $Q$ is the quantization function.

Mathematically, we express the quantized Smooth-SwiGLU function for each channel $i$ as:

$$\text{Smooth-SwiGLU}_{\hat{\mathbf{w}}_{1,i}, \hat{\mathbf{w}}_{2,i}}(\mathbf{x}) = s_i^{-1} \cdot Q(s_i \cdot (\hat{\mathbf{w}}_{1,i}^\top Q(\mathbf{x})) \text{Swish}(\hat{\mathbf{w}}_{2,i}^\top Q(\mathbf{x})))) \tag{3}$$

where $s_i$ is the per-channel scaling factor, $\hat{\mathbf{w}} = Q(\mathbf{w})$ denotes the quantized version of a weight vector $\mathbf{w}$, and $Q$ is a quantization function (in a slight abuse of notation, we suppress the dependence of $Q$ on the tensor).

To minimize computational overhead, we compute the scaling factors $s_i$ using an efficient parallel approach:

1. Split the tensor into chunks, where each chunk corresponds to a channel.

2. For each chunk (channel), compute its maximum value in parallel.

3. Use these per-channel maximum values to determine the individual scaling factors $s_i$ for each channel.

This method allows for efficient per-channel scaling, as each channel's scaling factor is computed independently and in parallel. The computational cost of this approach is moderate compared to the matrix multiplications in the linear layers, especially given the parallelization, even with our non optimized implementation. During inference, the scaling factors can be merged into the weights of the first and third linear layers in the MLP layer that includes the SwiGLU layer followed by a linear layer (see Figure 4), i.e.

$$\sum_i \hat{\mathbf{w}}_{3,i} \text{Smooth-SwiGLU}_{\hat{\mathbf{w}}_{1,i}, \hat{\mathbf{w}}_{2,i}}(\mathbf{x}) = \sum_i s_i^{-1} \cdot \hat{\mathbf{w}}_{3,i} Q(s_i \cdot (\hat{\mathbf{w}}_{1,i}^\top Q(\mathbf{x})) \text{Swish}(\hat{\mathbf{w}}_{2,i}^\top Q(\mathbf{x}))))$$

So we can absorb the scalar by re-defining $\tilde{\mathbf{w}}_{1,i} \triangleq Q(s_i \cdot \mathbf{w}_{1,i})$ and $\tilde{\mathbf{w}}_{3,i} \triangleq Q(s_i^{-1} \cdot \mathbf{w}_{3,i})$. Thus, this procedure results in zero additional cost at inference.

## 5  FP8 OPTIMIZER

The Adam optimizer and its variants are widely used in deep learning due to their effectiveness in handling various training challenges. A key characteristic of the Adam optimizer is its storage of two moments, traditionally in high precision (FP32). This significantly increases memory usage, particularly for large-scale models. While previous research Peng et al. (2023) has shown the feasibility of reducing the first moment to FP8 precision, they retained the second moment in FP16. Our work pushes the boundaries further by successfully quantizing both moments to FP8, significantly improving the optimizer efficiency for large language models.

## 5.1 CHALLENGES

The Adam optimizer uses two moments to adapt learning rates for each parameter:

1. The first moment is an estimate of the mean of the gradients.
2. The second moment is an estimate of the uncentered variance of the gradients.

A critical aspect of the Adam optimizer is the use of the inverse square root of the second moment in the parameter update step. This operation has important implications for precision requirements.

Due to this inverse square root operation, the smallest values in the second moment become the most significant in determining parameter updates. This characteristic creates a unique challenge when considering precision reduction for the second moment.

## 5.2 METHODOLOGY

We conducted extensive experiments to determine the optimal FP8 formats for both moments. Our investigation revealed that different precision requirements exist for each moment:

1. **First Moment**: The E4M3 format (4 exponent bits, 3 mantissa bits) provides sufficient precision. This format offers a good balance between range and accuracy for representing the mean of the gradients.
2. **Second Moment**: The E5M2 format (5 exponent bits, 2 mantissa bits) is necessary. This format provides a higher dynamic range, crucial for preserving information about the smallest values in the second moment. The additional exponent bit ensures that we can accurately represent both very small and moderately large values, which is critical given the inverse square root operation applied to this moment.

In our experiments, presented in Fig. 5 we show that while the first moment is able to converge with E4M3, the second moment, which estimates the square of the gradients, requires a wider dynamic range and is able to converge only with E5M2 format. In Table 1 we compare the proposed quantization scheme for the optimizer moments with the one presented in Peng et al. (2023). Our scheme shows, for the first time, the ability to quantize both moments with standard FP8 formats.

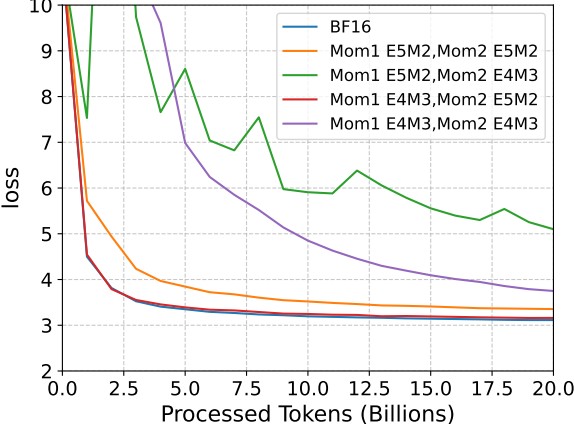

Figure 5: All combinations for quantization the Adam moments with standard FP8 formats in Llama2 100m. The only combination that is able to converge to baseline is first moment E4M3 format and second moment E5M2 format.

## 6 EXPERIMENTS

We conducted extensive experiments to evaluate the effectiveness of our proposed FP8 training method for Large Language Models (LLMs) across various scales.

Table 1: Comparison of the different datatypes for the two Adam optimizer moments.

| Model | Mom 1 | Mom 2 |
|---|---|---|
| BF16 (Baseline) | FP32 | FP32 |
| FP8 (Peng et al., 2023) | FP8 | FP16 |
| FP8 (Ours) | FP8 | FP8 |

## 6.1 EXPERIMENTAL SETUP

**Model and Dataset.** We used the Llama2 model (Touvron et al., 2023) as our baseline. This model is a decoder-only Transformer (Brown et al., 2020) with pre-normalization RMSNorm (Zhang & Sennrich, 2019), SwiGLU activation function (Shazeer, 2020b), and rotary positional embeddings (Su et al., 2024). We trained the models on the open-source Red Pajama dataset (Computer, 2023) for 2 trillion tokens, maintaining hyperparameters consistent with Touvron et al. (2023).

**Hardware.** All training was conducted on 256 Intel Gaudi2 devices.

## 6.2 RESULTS

**Training Stability.** In Fig. 6 we show the training loss of Llama2 with the proposed scheme, which includes the use of Smooth SwiGLU (Section 4.4) + FP8 quantization of both Adam moments (Section 5). Notice we can overcome the divergence point of standard FP8 training. The FP8 model was trained using the standard format (Micikevicius et al., 2022) which includes saving a high precision weight matrix and quantization to E4M3 for the forward phase and E5M2 for the backward phase with delayed scaling, similar to Nvidia's transformer Engine. The model was trained on 256 Intel Gaudi2 over 15 days.

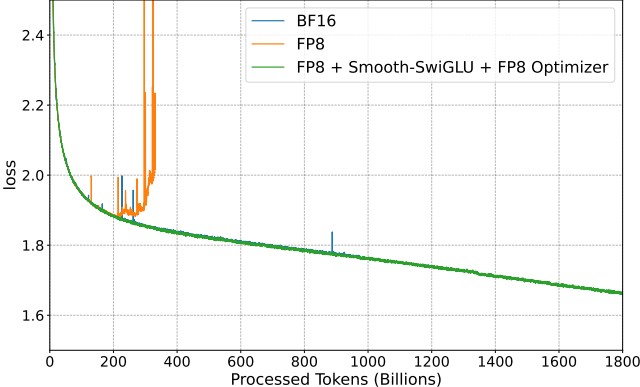

Figure 6: Training loss of Llama2 7B using the proposed Smooth SwiGLU as the activation function (Section 4.4) and with FP8 quantization of both moments of Adam optimizer (Section 5). As can be seen, the proposed scheme can converge similarly to the BF16 baseline while standard FP8 with SwiGLU and FP32 Adam moments diverge after 200B tokens.

**Zero-shot Performance.** Table 2 compares the zero-shot performance (accuracy and perplexity) on downstream tasks between the BF16 baseline and our FP8 model. The results demonstrate that our FP8 approach achieves on-par performance with the BF16 baseline across all tested metrics.

**Performance gains.** Table 3 presents the performance of different configurations on Intel Gaudi2 hardware. While full FP8 quantization achieves the highest acceleration ($\sim$37%), it leads to training divergence (as shown in Figure 2a). Disabling quantization for the $\mathbf{w}_3$ layer enables convergence (Figure 3) with a $\sim$27% speedup. Our proposed Smooth-SwiGLU scheme not only converges with results on par with the BF16 baseline (Figure 6) but also delivers a substantial $\sim$34% acceleration.

Table 2: Zero shot accuracy and perplexity comparison between the BF16 baseline and the proposed FP8. Notice both models achieve on-par results across all tests. FP8(1) refers to FP8 + SwiGLU output in BF16 while FP8(2) refers to FP8 + Smooth SwiGLU + FP8 optimizer.

| Precision | Accuracy ↑ | | | | Perplexity ↓ | |
|---|---|---|---|---|---|---|
| | Lambada | HellaSwag | Winogrande | Arc-C | Wikitext | Lambada |
| BF16 | 61.98 | 68.3 | 64.25 | 37.37 | 5.59 | 5.75 |
| FP8 (1) | 61.73 | 68.03 | 64.4 | 37.2 | 5.56 | 5.89 |
| FP8 (2) | 62.1 | 68.37 | 65.43 | 37.8 | 5.55 | 5.9 |

Table 3: Performance acceleration with different configurations in our non optimized implementation in Llama2 7B model. The measurement were done on 8 Intel Gaudi2 devices.

| Configuration | Micro BS | Status | Throughput (Samples/sec) | TFLOPS |
|---|---|---|---|---|
| BF16 | 1 | | 12.65 | 311 |
| FP8 + SwiGLU output in BF16 | 1 | Converge | 16.07 (+ 27.04 %) | 396 |
| FP8 + Smooth SwiGLU | 1 | Converge | 16.89 (+33.52 %) | 417 |
| FP8 | 1 | Diverge | 17.34 (+37.08 %) | 428 |

**Memory reduction.** In Table 4 we present the memory reduction achieved by changing the optimizer moments from standard FP32 to FP8. Moreover, we reduce the master weight to FP16, as shown in Peng et al. (2023). As can be seen, we can reduce the memory consumption by $\sim 30\%$,

Table 4: Memory reduction when applying the proposed FP8 optimizer (Section 5). The measurements were done on 8 Intel Gaudi2 devices, using Deepspeed Zero-1.

| Configuration | Status | Memory (GB/HPU) | FP8 Optimizer |
|---|---|---|---|
| BF16 | | 63.25 | ✗ |
| FP8 + SwiGLU output in BF16 | Converge | 63.26 | ✗ |
| FP8 + Smooth SwiGLU | Converge | 63.26 | ✗ |
| FP8 | Diverge | 63.24 | ✗ |
| FP8 + SwiGLU output in BF16 | Converge | 44.08 | ✓ |
| FP8 + Smooth SwiGLU | Converge | 44.08 | ✓ |
| FP8 | Diverge | 44.09 | ✓ |

## 7 CONCLUSIONS

In this paper, we successfully demonstrated FP8 training on datasets up to 2 trillion tokens, significantly exceeding the previous limit of 100 billion tokens Peng et al. (2023), with on-par results with the BF16 baseline. Importantly, we discovered that earlier FP8 training attempts were not long enough to reveal critical instabilities caused by outliers. Through both analytical methods and simulations, we showed that these outliers emerge over time, particularly in extended training runs. Our investigation revealed that the SwiGLU activation function amplifies these outliers, destabilizing FP8 training in large-scale scenarios.

To address this issue, we applied per-channel quantization to the SwiGLU activation function, a technique we refer to as Smooth-SwiGLU. Although identical to SwiGLU in function, this method effectively reduces outlier amplification, ensuring stable FP8 training with a moderate effect on model performance during training, and without any effect on the inference. Additionally, we introduced the first implementation of FP8 quantization for both Adam optimizer moments, further optimizing memory usage.

Our proposed method, combining Smooth-SwiGLU and FP8 optimizer moments, achieved comparable performance to BF16 baselines on downstream tasks while providing significant throughput improvements. This approach successfully overcome the divergence challenges typically encountered in standard FP8 training on large datasets.

**Reproducibility**  The abstract of the paper provides a link to an anonymous GitHub repository (`https://github.com/Anonymous1252022/Megatron-DeepSpeed`) containing all the code and necessary details for reproducing the experiments.

**Ethics**  LLMs require immense computational resources during training, which contributes significantly to carbon emissions. This environmental cost has become a growing concern in the field of AI. The use of low-precision formats like FP8 offers a promising solution, as it significantly reduces the computational overhead without sacrificing model accuracy. By adopting FP8 for training, not only can we enhance training efficiency, but we can also mitigate the carbon footprint associated with large-scale LLM training, paving the way for more sustainable AI development.

ACKNOWLEDGEMENTS

The research of DS was Funded by the European Union (ERC, A-B-C-Deep, 101039436). Views and opinions expressed are however those of the author only and do not necessarily reflect those of the European Union or the European Research Council Executive Agency (ERCEA). Neither the European Union nor the granting authority can be held responsible for them. DS also acknowledges the support of the Schmidt Career Advancement Chair in AI.

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
