# A APPENDIX

## A.1 TRAINING INSTABILITY - ADDITIONAL DATA

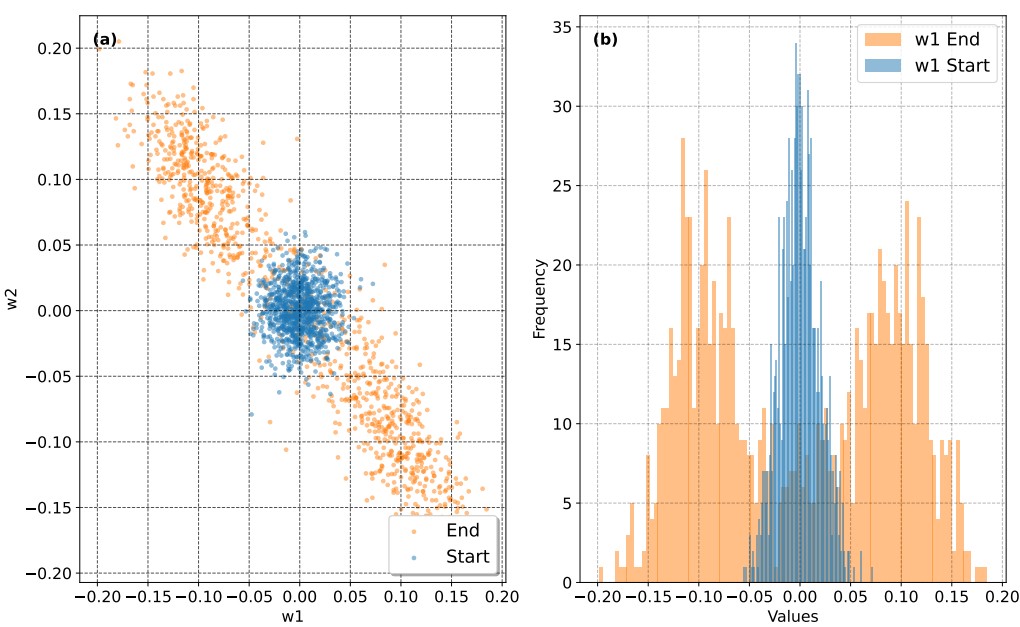

Figure 7: **(a):** Scatter plot of outlier channel elements in $\mathbf{w}_1$ and $\mathbf{w}_2$, at an early training stage (8B tokens) and late training stage (330B tokens), demonstrating minimal correlation at start of the training and high negative correlation in the later stage. **(b):** Histogram of an outlier channel with negative correlation of $\mathbf{w}_1$ at an early training stage (8B tokens) and late training stage (330B tokens).

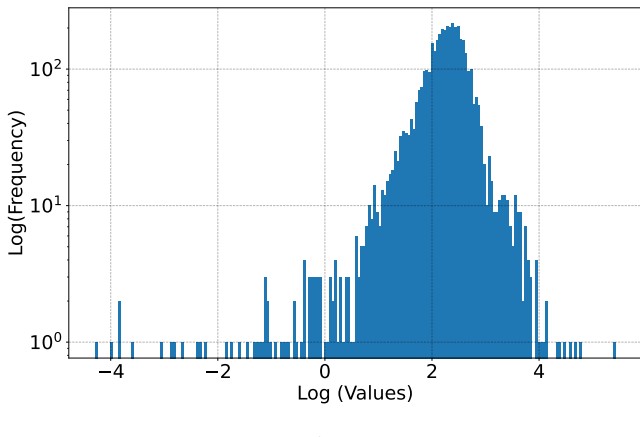

Figure 8

Figure 9: Histogram of $|\mathbf{w}_2^\top \mathbf{x}_n|$ for all the tokens in a single minibatch, at an outlier channel after training on 200B tokens. The x-axis is in natural (base-$e$) log scale, while the y-axis is base-10 log scale. We find that $\sim 1\%$ of the values are $< 1$ (0 in log scale) and $\sim 3.5\%$ of the values are $< e$ (1 in log scale). This implies that $\sigma'(\mathbf{w}_2^\top \mathbf{x}_n)$ is very small for the overwhelming majority of $n$ values.

## A.2 PERFORMANCE GAIN ON NVIDIA GPUs

In Table 5 we extend the perfomance acceleration comparison of Table 3 also for Nvidia GPUs.

Table 5: Performance acceleration with different configurations in our non optimized implementation in Llama2 7B model. The measurement were done on 8 Nvidia GPU A6000 Ada

| Configuration | Micro BS | Status | Throughput (Samples/sec) | TFLOPS |
|---|---|---|---|---|
| BF16 | 1 | | 3.22 | 76 |
| FP8 + SwiGLU output in BF16 | 1 | Converge | 4.11 (+ 27.6 %) | 96.9 |
| FP8 + Smooth SwiGLU | 1 | Converge | 4.32 (+34.16 %) | 101.9 |
| FP8 | 1 | Diverge | 4.43 (+37.58 %) | 104.5 |

## A.3 SMOOTH-SWIGLU STUDY

In Fig. 10 we show a study of the effect of Smooth-SwiGLU on BF16 training with different LR. Smooth-SwiGLU allows a smoother training curve even with BF16 training. Moreover, it enables training to lower loss values, especially when using a larger LR. (Fig. 11)

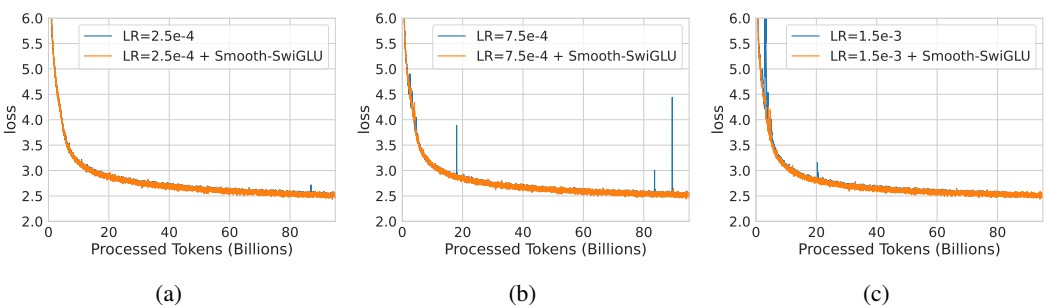

|     |     |     |
|:---:|:---:|:---:|
| (a) | (b) | (c) |

Figure 10: Effect of Smooth-SwiGLU on Llama 700m BF16 training. The LR refers to peak LR used with a standard cosine scheduler, where 2.5e-4 is the standard baseline. Notice Smooth-SwiGLU allows smoother training even with BF16 datatype.

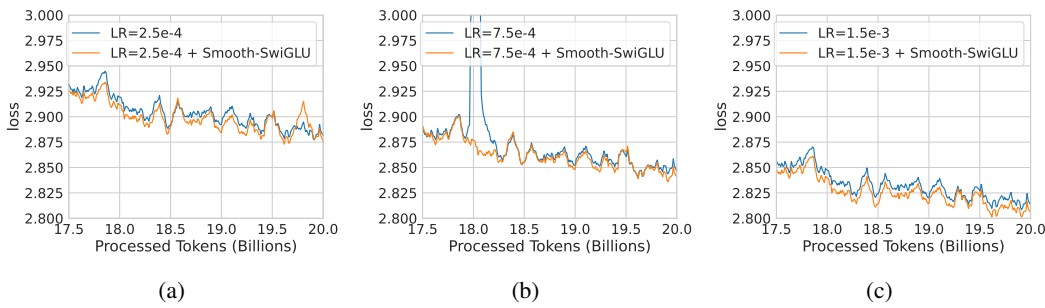

|     |     |     |
|:---:|:---:|:---:|
| (a) | (b) | (c) |

Figure 11: Zoom in of the effect of Smooth-SwiGLU on Llama 700m BF16 training(Fig. 10). Notice Smooth-SwiGLU allow to get lower loss.

## A.4 FP8 WITHOUT SWIGLU ACTIVATION FUNCTION

In Fig. 12 we show FP8 training on c4 dataset of GPT3 model, which include GeLU activation function. Notice, in this scenario no training stability issues were observed.

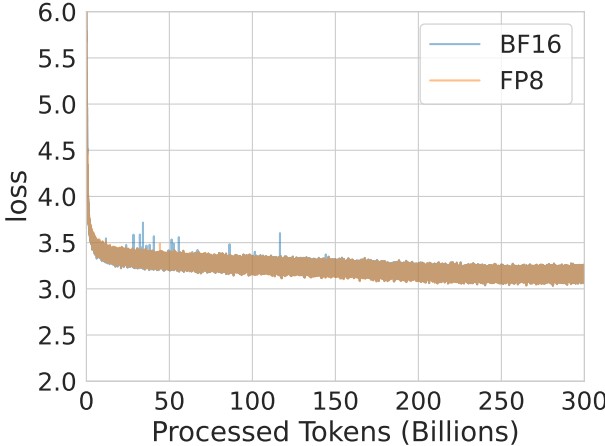

Figure 12: The FP8 training of the GPT-3 125M model demonstrated convergence of the training loss. It is important to note that this configuration did not incorporate the SwiGLU activation function.