# OpenReview forum: "Scaling FP8 training to trillion-token LLMs"
_ICLR.cc/2025/Conference — ICLR 2025 Spotlight_

### Official Review · Reviewer_KJr6 · 2024-10-26

**Soundness:** 3
**Presentation:** 4
**Contribution:** 3
**Rating:** 8
**Confidence:** 2

**Summary:**

This paper 1) identifies instability issues in FP8 training for large language models (LLMs) on trillion-token-scale datasets, 2) links the root cause to the alignment of two weight matrices of the SwiGLU neuron, and 3) proposes Smooth SwiGLU to mitigate outliers in the activations. Smooth SwiGLU achieves this by applying scaling factors before and after the last linear layer of the MLP components. Furthermore, they compared different FP8 formats to obtain the best configuration for the FP8 optimizer. The method successfully stabilizes FP8 LLM training on trillion-token-scale datasets and delivers a 5.1% throughput improvement over the baseline on 256 Intel Gaudi2 accelerators.

**Strengths:**

1. The paper uses comprehensive experiments and analysis to link the instabilities in FP8 training to SwiGLU weight alignment.
2. The paper provides an efficient implementation of Smooth-SwiGLU.
3. The paper is well-written.

**Weaknesses:**

1. The motivation of Smooth SwiGLU is that "While disabling quantization of the SwiGLU output effectively prevents divergence, it reduces the potential acceleration benefits of FP8". However, "FP8 + Smooth SwiGLU" is only 5.1% faster than the "FP8 + SwiGLU output BF16". Given that the performance gap is relatively small and the experiments were conducted only on Intel Gaudi2, it would be helpful to compare the two variants on more hardware configurations (i.e., NVIDIA and AMD GPUs) to verify whether the performance gap persists.
2. Table 2 currently only compares the accuracy and perplexity between the proposed FP8 and the standard BF16 configurations. To provide a more comprehensive evaluation, it would be beneficial to include additional comparisons, such as 'FP8 + Smooth SwiGLU,' 'FP8 + SwiGLU output BF16,' and 'FP8 + SwiGLU output BF16 + FP8 Optimizer'. If evaluating these variants is challenging or unnecessary, please provide the corresponding discussion.

**Questions:**

When training larger models, do similar issues arise earlier or later?

---

> ### Author Response · Authors · 2024-11-24
> **Response to reviewer KJr6**
>
> $\textbf{Q1}$: "The motivation of Smooth SwiGLU is that "While disabling quantization of the SwiGLU output effectively prevents divergence, it reduces the potential acceleration benefits of FP8". However, "FP8 + Smooth SwiGLU" is only 5.1\% faster than the "FP8 + SwiGLU output BF16". Given that the performance gap is relatively small and the experiments were conducted only on Intel Gaudi2, it would be helpful to compare the two variants on more hardware configurations (i.e., NVIDIA and AMD GPUs) to verify whether the performance gap persists."
>
> $\textbf{A1}$: Following the reviewer's request, we added in the Appendix (Section A.2) performance acceleration with the different configurations on Nvidia GPUs (A6000 Ada). Notice, the performance acceleration ratio is similar for Gaudi2 Vs Nvidia: 27.04\% Vs 27.6\% for 'FP8 + SwiGLU output in BF16', 33.52\% Vs 34.16\% for 'FP8 + Smooth SwiGLU' and 37.08\% Vs 37.58\% for 'FP8'. Unfortunately, we do not have access to AMD GPUs or H100 GPUs (to show better performance).
>
> $\textbf{Q2}$: "Table 2 currently only compares the accuracy and perplexity between the proposed FP8 and the standard BF16 configurations. To provide a more comprehensive evaluation, it would be beneficial to include additional comparisons, such as 'FP8 + Smooth SwiGLU,' 'FP8 + SwiGLU output BF16,' and 'FP8 + SwiGLU output BF16 + FP8 Optimizer'. If evaluating these variants is challenging or unnecessary, please provide the corresponding discussion."
>
> $\textbf{A2}$: We thank the reviewer for pointing this out, we missed this. We updated Table 2 to contain the 2 options we run: 'FP8 + SwiGLU output in BF16' (Fig 3) and 'FP8 + Smooth-SwiGLU + FP8 optimizer' (Fig 6). Unfortunately, we didn't run the other options: 'FP8 + SwiGLU output in BF16 + FP8 optimizer' or 'FP8 + Smooth-SwiGLU', since each run takes more than 2 weeks on 256 devices (see `general comment' above). However, we expect similar accuracy since from our experiments the FP8 optimizer converges similarly to the full precision counterpart.
>
> $\textbf{Q3}$: "When training larger models, do similar issues arise earlier or later?"
>
> $\textbf{A3}$: Excellent question! Our analysis in section 4.2 is around a stationary point, so the alignment effect should become stronger as we converge closer to the stationary point. Also, as the reviewer can notice from Fig. 5 in Llama2 paper [1], larger models often achieve a smaller loss in fewer steps. Combining both observations, we believe that, in larger models, the alignment effect (and its resulting outliers) should appear earlier during training (in terms of steps). Of course, this is something that we need to verify empirically --- and we plan to do so when we will have enough resources.
>
> =============================
>
> [1] Llama2 - https://arxiv.org/pdf/2307.09288

---

### Official Review · Reviewer_1QzP · 2024-10-27

**Soundness:** 3
**Presentation:** 3
**Contribution:** 2
**Rating:** 6
**Confidence:** 3

**Summary:**

This paper proposed a practical solution for quantized (FP8) training over trillion tokens.

The paper conduct LLM training over a real big 2 trillion dataset and found swiglu is the main cause of instability when training using FP8, thus proposed a optimized and smoothed method called smooth-swiglu.

Smooth-swiglu in together with FP8, the authors achieve good model convergence on 2 trillion token dataset for llm training

**Strengths:**

Training LLM on trillion-token big dataset makes the paper result solid.

Adopting a pure FP8 adam optimizer (both two momentum are fp8) and make training converge is good contribution.

Identify swiglu cause model training instability and optimized as a smooth-swiglu is another contribution.

**Weaknesses:**

1 ) Paper novelty is limited. The only thing new is adding a scaling factor on top of swiglu may not be a big novelty. Previous work already study similar smooth method on activation functions. e.g., Swish, SMU as smooth function for relu.

[1] swish: https://arxiv.org/pdf/1710.05941

[2] SMU: https://arxiv.org/pdf/2111.04682

2 ) FP8 optimizer contribution is just extending previous 1 momentum using FP8 to both momentum using FP8 may lack a bit novelty here. For example, Figure 5 shows empirical study for both FP8 momentum on llama2 100m, it shows second momentum using E5M2 format achieve similar model loss curve as bf16. But how does it generalize to bigger models (more practical models like llama 8b 70b)? Is second momentum always need larger dynamic range rather than higher precision?

3 ) The only baseline is from this paper[3], how does this approach compared with nvidia transformer engine fp8. There is no comparison in either design or evaluation results. Please include a comparison with NVIDIA's Transformer Engine FP8 implementation, both in terms of methodology and empirical results. This comparison would provide valuable context for the paper's contributions.

[3]FP8-LM: Training FP8 Large Language Models

4 ) Billion level token training is almost sufficient for most LLM downstream tasks (e.g. Apple [4]). This trillion-token improvement may not have wide application scenarios. Please discuss more on potential use cases where trillion-token training might be beneficial, and how this paper's method scales compared to existing approaches at different dataset sizes.

[4] Apple Intelligence Foundation Language Models https://arxiv.org/pdf/2407.21075

**Questions:**

How does this paper compare with nvidia transformer-engine's fp8 training with automatic mixed precision?

In table 3, why micro batch is 1? The reason for doing quantization is to support larger batch training (higher throughput), if limiting to micro-batch as 1, it is almost impossible to get good throughput/token per sec numbers.

In table 4, deepspeed zero-1 itself would reducing some GPU memory footprint compared with pure pytorch DDP. Why use zero1 not 2 or 3?

In figure 6, why bf16 loss curve has more spikes compared with FP8 + Smooth-SwiGLU + FP8 Optimizer? to me, I think bf16 should have better and more smooth loss curve compared with any FP8 methods?

---

> ### Author Response · Authors · 2024-11-24
> **Response to reviewer 1QzP 1/2**
>
> $\textbf{Q1}$: " Paper novelty is limited. The only thing new is adding a scaling factor on top of swiglu may not be a big novelty. Previous work already study similar smooth method on activation functions. e.g., Swish, SMU as smooth function for relu. "
>
> $\textbf{A1}$: There may have been a misunderstanding here: Smooth-SwiGLU’s scaling factors do not smooth out the activations (as in Swish and SMU), as they appear before (and after) the quantization function (see Fig. 4), not before the activation. Note that without the quantization function, these scaling factors will cancel out. This scaling ensures stability for FP8 training without modifying SwiGLU's functional properties and with minimal implementation complexity. Perhaps our choice of the name `Smooth SwiGLU' was not optimal and led to this misunderstanding.
>
> Moreover, please note our paper has several other novel contributions besides the smooth-SwiGLU (listed in the bullet list in the introduction section), such as the first FP8 training on a trillion-token dataset and pinpointing the FP8 stability issue to SwiGLU's weight alignment problem.
>
> $\textbf{Q2}$: "FP8 optimizer contribution is just extending previous 1 momentum using FP8 to both momentum using FP8 may lack a bit novelty here. For example, Figure 5 shows empirical study for both FP8 momentum on llama2 100m, it shows second momentum using E5M2 format achieve similar model loss curve as bf16. But how does it generalize to bigger models (more practical models like llama 8b 70b)? Is second momentum always need larger dynamic range rather than higher precision? "
>
> $\textbf{A2}$: Please note the result in Fig. 6: there we show we can train a 7B model also with an FP8 optimizer (first moment E4M3, second moment E5M2) --- i.e. we show the generalization of the FP8 optimizer recipe also for a practical model, as the reviewer required.
> As noted in "general comments", training Llama2 7B takes about 2 weeks in 256 devices. Training a 70B model would require even more computational resources and time, which we do not currently have.
> However, we indeed believe that the second moment would generally require a higher dynamic range than the first moment --- as it is natural to have a significantly larger range for the (estimator of the) square of the gradients than for the (estimator of the) gradients. We have clarified this point in the revised version of the manuscript. Thank you for bringing it to our attention.
>
> $\textbf{Q3}$: "The only baseline is from this paper [3], how does this approach compared with nvidia transformer engine fp8. There is no comparison in either design or evaluation results. Please include a comparison with NVIDIA's Transformer Engine FP8 implementation, both in terms of methodology and empirical results. This comparison would provide valuable context for the paper's contributions. "
>
> $\textbf{A3}$: Please note that both [3] and the Nvidia transformer engine use the same quantization configuration which includes delayed scaling, the E4M3 format for the forward phase, and the E5M2 format for the backward phase. In the paper, we use Gaudi's implementation for the transformer engine which is equivalent. We clarify this point in the experiment section (line 454). Unfortunately, there are no open-source models that were trained on FP8 with Nvidia's transformer engine with similar configurations, and running it requires extensive GPUs resources, which we do not have. Therefore, we cannot run FP8 training with Nvidia's transformer engine directly.
>
> $\textbf{Q4}$: "Billion level token training is almost sufficient for most LLM downstream tasks (e.g. Apple [4]). This trillion-token improvement may not have wide application scenarios. Please discuss more on potential use cases where trillion-token training might be beneficial, and how this paper's method scales compared to existing approaches at different dataset sizes.
>
> $\textbf{A4}$: Perhaps this is a misunderstanding: our paper refers to trillions of tokens, but the models can still have billions of parameters. Please note models in [4] are also trained for trillions of tokens. For example, in section 3.2.1, the `AFM-server' paragraph in [4]: "We train AFM server from scratch for 6.3T tokens...",  and in the AFM-on-device paragraph "training for a full 6.3T tokens".
> Moreover, one can find many modern foundation models that were trained for trillions of tokens, such as llama2 (2T tokens),  llama3 (15T tokens), and mistral (8T tokens).
> Thus, we believe trillion tokens training is a common real-world scenario. In this work, we show for the first time the ability to train in this scenario with FP8 precision.

---

> ### Author Response · Authors · 2024-11-24
> **Response to reviewer 1QzP 2/2**
>
> $\textbf{Q5}$: "In table 3, why micro batch is 1? The reason for doing quantization is to support larger batch training (higher throughput), if limiting to micro-batch as 1, it is almost impossible to get good throughput/token per sec numbers."
>
> $\textbf{A5}$: The main purpose of this work is to show the ability to accelerate compute by using FP8 matrix multiplications instead of BF16. We use the standard recipe of low precision training (similar to [3] and Nvidia transformer engine), which includes storing the high-precision weights, and only quantizing the weights right before the matrix multiplication (done at the forward/backward phases). In general, this recipe does not allow using larger batch sizes than the BF16 training regime (which used a batch size of 1) since it does not reduce memory. We clarify the specific quantization recipe in the updated version of the paper in experiments section (line 454).
>
> $\textbf{Q6}$: "In table 4, deepspeed zero-1 itself would reducing some GPU memory footprint compared with pure pytorch DDP. Why use zero1 not 2 or 3?"
>
> $\textbf{A6}$: We agree with the reviewer that using Zero2 or Zero3 can allow additional memory reduction --- however we decided to use Zero1 since it is the method that reduces memory with sharding of the optimizer moments, that are the focus in table 4 and allow a simpler implementation.  Notice also the BF16 experiments in table 4 use Zero1, so it is a fair comparison with the proposed method.
>
> $\textbf{Q7}$: "In figure 6, why bf16 loss curve has more spikes compared with FP8 + Smooth-SwiGLU + FP8 Optimizer? to me, I think bf16 should have better and more smooth loss curve compared with any FP8 methods?"
>
> $\textbf{A7}$: Excellent question! It was previously observed that the BF16 loss curve can have spikes (e.g. figure 4 in [5]). Indeed, as the reviewer pointed out, it seems that Smooth-SwiGLU has strong stabilizing properties --- so strong that FP8 training with Smooth-SwiGLU is more stable (less `spiky') than BF16 training. Following the reviewer's question, to further examine this stabilizing effect, we added to the appendix (Section A.3), a study on the effect of Smooth-SwiGLU on BF16 training. As we also mention in the ``general comment'', we find two additional interesting conclusions: (1) Smooth-SwiGLU allows a smoother training curve using standard LR. (2) Smooth-SwiGLU us get to lower loss values, especially with larger LR. So indeed, Smooth-SwiGLU has a significant stabilizing effect also in BF16 training. In future work, we plan to investigate this direction on a large scale.
>
> ===================================
>
> [1] swish: https://arxiv.org/pdf/1710.05941
>
> [2] SMU: https://arxiv.org/pdf/2111.04682
>
> [3] FP8-LM: Training FP8 Large Language Models
>
> [4] Apple Intelligence Foundation Language Models https://arxiv.org/pdf/2407.21075"
>
> [5] Micro scaling: https://arxiv.org/pdf/2310.10537

---

> > ### Comment · Reviewer_1QzP · 2024-11-29
> > **official comments by reviewer 1QzP**
> >
> > Thank authors for the reply, which addressed my major concerns. I have updated my score, and I think this is a solid paper.

---

### Official Review · Reviewer_p6vg · 2024-11-02

**Soundness:** 3
**Presentation:** 3
**Contribution:** 3
**Rating:** 8
**Confidence:** 3

**Summary:**

The paper presents a novel approach to training large language models (LLMs) using FP8 precision on datasets of up to 2 trillion tokens, revealing critical instabilities associated with FP8 training that were not observable in prior studies. It identifies that the SwiGLU activation function amplifies outliers over prolonged training, leading to instability. To mitigate this, the authors propose Smooth-SwiGLU, a modified activation function that maintains stability without altering performance. The paper also introduces FP8 quantization for both Adam optimizer moments, significantly improving memory usage and training throughput.

**Strengths:**

Very interesting findings!

The introduction of Smooth-SwiGLU and the quantization of Adam optimizer moments are significant advancements in the FP8 training methodology.

The paper provides thorough experimental results demonstrating the effectiveness of the proposed techniques, achieving comparable performance to BF16 with improved training throughput.

Successfully training models on datasets up to 2 trillion tokens sets a new benchmark for FP8 training, addressing scalability issues in LLMs.

**Weaknesses:**

While the findings are significant, their applicability to other model architectures beyond those tested (like LLaMA2) could be explored further.

**Questions:**

How do the results generalize to other activation functions beyond SwiGLU?

---

> ### Author Response · Authors · 2024-11-24
> **Response to reviewer p6vg**
>
> $\textbf{Q1}$: "While the findings are significant, their applicability to other model architectures beyond those tested (like LLaMA2) could be explored further... How do the results generalize to other activation functions beyond SwiGLU?"
>
> $\textbf{A1}$: According to our analysis in section 4.2, replacing the Swish activation in SwiGLU with any other GLU activation function ([1]) will suffer from similar phenomena, leading to alignment and quadratic amplification. This is in contrast to other activation functions (e.g., ReLU,GeLU) which are linear at larger input magnitudes. We added this clarification in the new version of the paper. To validate the analysis, we added Section A.4 in the appendix, detailing FP8 training of a GPT-3 125m model using the GeLU activation function, for standard dataset length. The results demonstrate that no training stability issues were observed in this scenario.
>
>
> [1] GLU Variants: https://arxiv.org/pdf/2002.05202

---

> > ### Comment · Reviewer_p6vg · 2024-11-30
> >
> > Thanks for the reply

---

### Official Review · Reviewer_8YGg · 2024-11-04

**Soundness:** 3
**Presentation:** 3
**Contribution:** 3
**Rating:** 8
**Confidence:** 4

**Summary:**

The paper presents new findings and addresses key challenges in large-scale FP8 training for LLMs on modern hardware that supports FP8 operations. The authors identified that one major challenge in scaling FP8 training is the occurrence of sporadic, high-magnitude activations. They pinpoint SwiGLU as the main source of these extreme outliers, providing the novel insight that a weight alignment process during training leads to substantial SwiGLU activations (as discussed in Section 4.1). To address this issue and stabilize large-scale FP8 training, the authors propose a smooth-SwiGLU activation function. This approach prevents outliers in the quantization of inputs to the last linear layer with an efficient per-channel scaling for better parallelism.

**Strengths:**

1. The paper demonstrates opportunities and challenges in scaling FP8 training to trillion-token scale using LLaMA2 architecture. In particular, the authors identify that with SwiGLU (which is an important to contemporary LLMs), weight alignment issues can induce large-magnitude outliers and result in training stability issue that poses significant challenge for FP8's limited dynamic range.

2. Extensive experiment results are provided, showing the proopsed Smooth-SwiGLU training with FP8 optimizer (moment 1: E4M3,moment 2: E4M3) is able to converge to baseline FP16 training.

3. Experiments show around 30% improvement in training throughput, which provides significant energy saving for large-scale training.

4. Paper writing and presentation are clear with well identified bottleneck and detailed solutions.

**Weaknesses:**

1. The paper has an extensive discussion about how feedforward SwiGLU affect FP8 training stability. However, there is no mention about how other model components like RMSNorm, MHA/GQA affect training stability. Could the authors also discuss whether each of the other model components affect FP8 training stability and provide quantitative results?

2. It seems the SwiGLU weight alignment issue explains the occurrence of huge-magnitude outliers. However, can the authors comment more on the sporadic nature of these outliers? Is SwiGLU held accountable for that as well?

**Questions:**

I would like to ask the authors for several clarifications:

1. If I understand correctly, smooth-SwiGLU does not directly address the weight alignment problem. Rather, it is an alternative implementation specifically designed for FP8 training, which circumvents the overflow issue by preventing outliers in the inputs to the last linear layer. Is this accurate?

2. Figure 2 (b) on the dynamics of $\mathbf w_1$ and $\mathbf w_2$ norm correlation is insightful and interesting. Is the dynamics based on llama2's trianing hypermaraters? Do the authors have additional empirical results on how these dynamics change with different training hyperparameters?

---

> ### Author Response · Authors · 2024-11-24
> **Response to reviewer 8YGg**
>
> $\textbf{Q1}$: "The paper has an extensive discussion about how feedforward SwiGLU affect FP8 training stability. However, there is no mention about how other model components like RMSNorm, MHA/GQA affect training stability. Could the authors also discuss whether each of the other model components affect FP8 training stability and provide quantitative results?"
>
> $\textbf{A1}$: Excellent question! As the reviewer can notice from Fig. 3 --- when we disable the FP8 quantization only at the SwiGLU output the loss converges similarly to the BF16 baseline. This means the other components, such as MHA or RMS Norm, do not cause significant instability with FP8 precision. Notice that GQA was not checked, since it is not part of Llama2 7B architecture. However, we plan in future work to run Llama3, where GQA is included. We added this clarification in the updated version.
>
> $\textbf{Q2}$: "It seems the SwiGLU weight alignment issue explains the occurrence of huge-magnitude outliers. However, can the authors comment more on the sporadic nature of these outliers? Is SwiGLU held accountable for that as well?"
>
> $\textbf{A2}$: To get these immense outliers several things need to occur --- weights alignments ($\mathbf{w}_1\sim \pm \mathbf{w}_2$), high norm weights ($||\mathbf{w}_i||\gg 1$), and specific tokens for which the layer input $\mathbf{x}$ is aligned with the layer weight vectors $\mathbf{w}_i$ (i.e., that $\mathbf{x}$ and $\mathbf{w}_i$ are not near-orthogonal). For these specific tokens, this causes high pre-activation values (i.e., that $|\mathbf{w}_i\cdot\mathbf{x}|\gg 1$), which can be amplified quadratically by the SwiGLU activation --- leading to instability with low-precision training.
>
> $\textbf{Q3}$: "If I understand correctly, smooth-SwiGLU does not directly address the weight alignment phenomenon. Rather, it is an alternative implementation specifically designed for FP8 training, which circumvents the overflow issue by preventing outliers in the inputs to the last linear layer. Is this accurate?"
>
> $\textbf{A3}$: Exactly, Smooth-SwiGLU doesn't prevent the alignment phenomenon, but it allows FP8 training even with this alignment. We did not try to prevent this alignment since in quantized training we aim to reduce numerical precision without significantly changing the original training regime. The reason FP8 training works now is because Smooth-SwiGLU seems to stabilize training in general, as mentioned in the ''general comment" on the new section (A.3) that includes a study of the effect of Smooth-SwiGLU on BF16 training.
>
> $\textbf{Q4}$: "Figure 2 (b) on the dynamics of w1
>  and w2 norm correlation is insightful and interesting. Is the dynamics based on llama2's trianing hypermaraters? Do the authors have additional empirical results on how these dynamics change with different training hyperparameters?"
>
> $\textbf{A4}$: Yes, we used llama2 default hyperparameters as remarked in section 6.1. We believe it is reasonable to focus only on the default parameters since (1) llama's default hyperparameters are tuned to achieve the best results and (2) the purpose of this work is to allow FP8 training without requiring any additional tuning --- which requires many computational resources. Moreover, please notice in ``general comments'', we remark that the training time was over 2 weeks with 256 devices, and this limits us in checking additional training hyperparameters.

---

> > ### Comment · Reviewer_8YGg · 2024-11-26
> >
> > Thanks for the informative responses. This is overall a well-executed and thoughtfully presented work.  I believe it makes a valuable contribution by advancing the understanding of large-scale FP8 training and showcasing a cost-effective technique for scaling it further.

---

### Author Response · Authors · 2024-11-24
**General comment**

We thank all the reviewers for their detailed feedback and for expressing positive opinions. We uploaded a new revision of the paper and supplementary material to address all remarks. All the changes are marked in red.

Notice that we now mention in the experiments section that $\textbf{each training run takes more than 2 weeks with 256 devices}$. This limits us from doing many additional experiments.

Moreover, in the Appendix (section A.3), we added an interesting study of the effect of Smooth-SwiGLU on BF16 training. We show two interesting conclusions: (1) Smooth-SwiGLU allows a smoother training curve using standard LR. (2) Smooth-SwiGLU enables us get a lower loss, especially with a larger LR. This suggests that Smooth-SwiGLU is beneficial for training stability in general, not only for FP8 training. We plan to investigate this direction on a large scale in future work.

For additional details, see the answer for each reviewer concerns. Please let us know if there are any additional comments.

---

### Author Response · Authors · 2024-12-01
**General comment**

We sincerely thank the reviewers for their time and support. This constructive feedback has helped us improve the clarity and depth of the paper. We are happy that we have addressed all major concerns and strengthened the overall quality.

---

> ### Public Comment · ~Clarence_Lee3 · 2024-12-04
> **Validating the relevancy of this work**
>
> Dear authors and reviewers,
>
> I am a member of the public not affiliated with the authors, but I have worked extensively with pretraining. The problem that the authors tackle is highly relevant and impactful, and I personally have faced huge instabilities in fp8 training beyond 200B tokens and there was no clear solution of how to tackle this problem. Given that the authors have clearly validated their work at trillion token scale, I would like to appreciate the authors for this impactful work and hope that the reviewers can take into account the practical impact of this work when giving a final assessment.
>
> Thank you!

---

> > ### Author Response · Authors · 2024-12-04
> > **Comment to Clarence Lee**
> >
> > Thank you for your thoughtful comment; we're delighted to hear that our work has been helpful to you.

---

### Meta-Review · Area_Chair_kXew · 2024-12-23

**Metareview:**

All reviewers agreed this paper should be accepted: it addresses an important problem, the method is thoughtfully-designed, and the paper is clearly written. A clear accept. Authors: you've already indicated that you've updated the submission to respond to reviewer changes, if you could double check their comments for any recommendation you may have missed on accident that would be great! The paper will make a great contribution to the conference!

**Additional Comments On Reviewer Discussion:**

Three reviewers responded to the author feedback with very short responses, one raised their score. No authors engaged in further discussion of the paper. All reviewers agreed to accept. Reviewer p6vg wrote an extremely short review, I disregarded it. I wouldn't recommend inviting them for future ICLR review cycles.

---

### Decision · Program_Chairs · 2025-01-22

Accept (Spotlight)